# The Extant Shore-Platform Stromatolite (SPS) Facies Association: A glimpse into the Archean?

Alan Smith[1,6], Andrew Cooper[1,2], Saumitra Misra[1], Vishal Bharuth[3], Lisa Guastella[4, 6], Riaan Botes[5]

[1] Discipline of Geology, School of Agriculture, Earth and Environmental Sciences, University of KwaZulu-Natal, Westville 3630, South Africa

[2] School of Environmental Science, University of Ulster, Coleraine, Northern Ireland, UK

[3] Microscopy and Microanalysis Unit, University of KwaZulu Natal, Westville 3630, South Africa

[4] Bayworld Centre for Research and Education (BCRE), Cape Town, South Africa.

[5] Geo-Dynamic Systems, PO Box 1283, Westville 3630, South Africa

[6] Coast Busters Inc Research Group, 29 Brown's Grove, Sherwood, Durban 4091, South Africa

*Correspondence to*: Alan Smith (asconsulting@telkomsa.net)

**Abstract.** Shore platform stromatolites (SPS) were first noted from Cape Morgan on the southeast African seaboard. Since then they have been found growing discontinuously in rocky peritidal zones along the entire southern African seaboard. In addition they have been found on the southwest Australian coast, Giant's Causeway, N. Ireland, and more recently at Harris on the Scottish Hebridean Atlantic coast. In this paper their occurrence and their potential as analogues for Precambrian fossil stromatolites and potential Martian occurrences is assessed. Sub-horizontal surfaces promote stromatolite development, while tufa develops on cliffs and steep rocky surfaces. Tufa and stromatolites are end members of a spectrum dictated by coastal topography. Extant SPS occur on well indurated shore platforms in high wave energy settings, often around or near headlands. They can be associated with boulder beaches, boulder ridges, storm swash-terraces, coastal dunes and peat bogs. In contrast to other extant stromatolites, SPS are produced primarily by mineral precipitation, although minor trapping and binding stromatolites do occur. From a geological perspective, SPS are developing in mildly transgressive siliciclastic settings in various climatic and tidal regimes. We suggest that SPS could be preserved in the geological record as micritic lenses on palaeo-shore platform surfaces. SPS share many features with Precambrian stromatolites and are a valid modern analogue despite the widely different atmospheric and oceanic conditions of the Archean. We suggest that terraces associated with former oceanic or lacustrine flooding surfaces on Mars are potential targets in the search for palaeo-SPS on Mars.

# 1 Introduction

The oldest known stromatolites include those of the Isua Group (3.7 Ga), Greenland (Nutman et al., 2016), the Strelley Pool occurrence (3.43 Ga), Australia (Allwood et al., 2006), the Barberton Mountain Land (3.22 Ga), South Africa (Gamper et al., 2011) and the Pongola Group (2.9 Ga), South Africa (Mason & von Bruun, 1977; Bolhar et al., 2015). Comparison of the environment in which extant and sub-fossil stromatolites occur with ancient examples may advance our knowledge concerning the conditions under which life developed and in what environment it began. Stromatolite-building organisms probably dominated the Earth during the Archean and Proterozoic Eons, but under contemporary conditions only thrive in extreme environments that limit Metozoan competition. Such environments include geothermal springs (Jones et al., 2000; Berelsen et al., 2011) peritidal marine environments (Logan et al., 1964; Reid et al., 2000; Smith & Uken, 2003; Smith et al., 2005; 2011; Perissinotto et al., 2014; Rishworth et al 2016; Edwards et al., 2017) and salt lakes (Martin & Wilczewski, 1972). Prokaryotes are also recorded from the Earth's upper crust to depths of (at least) 7km (Sankaran, 1997) and from within the atmosphere (DeLeon-Rodriguez et al., 2012).

Stromatolites are biosedimentary structures produced by sediment trapping and binding and/ or mineral precipitation as a result of the growth and metabolic activity of a microbial community (Awramik & Marguilis, 1976; Burne & Moore, 1987). The best known extant stromatolite models are based on the trapped and bound stromatolites of Shark Bay, Western Australia and several sites in the Caribbean (Logan et al., 1964; Reid & Browne, 1991). In contrast, Precambrian stromatolites are of the mineral precipitation variety (Awramik & Grey, 2005; Reid et al., 2011). Shore-platform stromatolites (SPS) are mineral-precipitated (Smith et al.2011; Perissinotto et al., 2014; Rishworth et al. 2016; Edwards et al., 2017) and offer a plausible Precambrian stromatolite analog and deserve scrutiny.

Cape Morgan (Fig. 1) in the Eastern Cape Province, South Africa, was the first location where SPS was found. They form on shore platforms within the peritidal zone where suitable (carbonate-rich) terrestrial runoff is present (Smith et al., 2011; Perissinotto et al., 2014; Rishworth et al., 2016). The supratidal/ high intertidal zone experiences extreme environmental changes, which partially excludes Metazoans (Perissinotto et al., 2014; Rishworth et al., 2016) and enables prokaryotes to dominate. The Cape Morgan locality was discovered by Mountain (1937), although its significance was only realized much later (Smith & Uken, 2003). Smith et al. (2011) proposed the extant Cape Morgan (Fig. 1) stromatolites as a partial analogue for the 3.43 Ga stromatolites from Strelley Pool Australia. Smith et al. (2011) documented extant and subfossil SPS from Cape Morgan and indicated that they were found in patches from Tofo (Mozambique) to Port Elizabeth (Eastern Cape, South Africa)(Fig. 1). Since then other colonies have been found in Northern Ireland (Cooper et al., 2013) and further localities in

the Eastern Cape (Perissinotto et al, 2014; Rishworth et al., 2016; Edwards et al., 2017), and elsewhere in southern Africa (Fig. 1). More recently new discoveries have been made on the west coast of Harris, in the Scottish Hebrides.

[Fig. 1]

# 3 Methodology

This work is heavily reliant on fieldwork which has taken advantage of many serendipitous trips. No attempt has been made to institute a scientific survey due to the distances involved and limited manpower. Stromatolite (SPS of this paper) morphology, microstructure and ecology is discussed in detail elsewhere (Smith et al., 2011; Perissinotto et al., 2014; Rishworth et al., 2016; 2017; Edwards et al., 2017). New localities are discussed and compared with known. Such review is required to establish similarities and differences. This paper will focus on the synthesis of a Facies Association using tabulated geomorphological, lithological, oceanographic and climatological elements. This is a relatively new branch of stromatolites science and is heavily reliant on description and comparisons.

We document all known SPS localities (Table 1) with the aim of distillation of a generalised global SPS Facies Association. To this end we also describe several new SPS localities both on the southern African eastern seaboard and from the Atlantic side of Harris Island in the Scottish Hebrides, UK. Then we compile and review the physical, oceanographic, climatological, lithological and geomorphological settings in which extant SPS are forming in the Scottish Hebrides, the northeast coast of Northern Ireland and the southern African seaboard. All known SPS location coordinates and source data are given in Table 1. On the basis of this review, we present a Facies Association model for SPS.

We then compare the SPS Facies Association to extant stromatolites from contrasting peritidal environments in Hamelin Pool, W. Australia (Logan et al., 1964), Kuwait (Alshuaibi et al 2015), the Bahamas (Reid et al., 2000) and the Dutch North Sea (Kremer et al., 2008). Further, on the basis of this comparison, extant SPS are assessed as potential analogues for Precambrian stromatolites. We also comment briefly on potential targets for SPS on Mars.

# 4 Results

4.1. SPS Distribution

Extant shore platform stromatolites (SPS) occur discontinuously along a 2 300 km stretch of the southern African eastern seaboard (Fig. 1; Table 1). In addition extant SPS have been documented from W. Australia (Forbes et al., 2010) and Giant's

Causeway, Northern Ireland (Cooper et al., 2013). Previously unreported occurrences in the UK are also documented here. All known SPS occurrences are based on opportunistic observation as no systematic survey of their location has yet been made. SPS often occurs in association with tufa but the proportions vary with the coastal geomorphology. All are fed by spring water emanating from the terrestrial hinterland (Smith et al., 2011; Perissinotto et al., 2014; Edwards et al., 2017). If the coastline is cliffed or dominated by high-angle surfaces, tufa dominates (Fig. 2A; B), whereas SPS typically occurs within rock pools on competent sub-horizontal shore-platforms (Fig. 2A). In many instances there is a clear spatial gradation from tufa to stromatolites (2A and B). Most SPS develops directly on the shore platform but the trapped and bound (Fig. 3) variety develops adjacent to, and in channels cut into, the shore platform.

[Fig.2]

4.2.  Trapped & Bound SPS

Trapped and bound stromatolites are relatively rare, but have only been observed at Cape Morgan (Smith et al., 2011; Fig. 3), Seaview Skoenmakerskop (South Africa: see Edwards et al., 2017) and the north coast of Luskentyre Bay (Harris, UK). This stromatolite type appears to be restricted to clastic environments associated with the shore platform system and hence are still termed SPS.  Trapped and bound stromatolites are found growing on cemented beach gravel in breaks and gullies in the shore platform and beaches adjacent to the shore platform. The beach grain calibres involved vary from fine sand to boulders. A storm beach deposit (6-10 cm thick) bound between two mineralized laminar stromatolites has been observed (Fig. 3B) (Smith et al., 2011). In addition, within a depression in the Cape Morgan shore platform, a headland conglomerate formed by stromatolite cementation of dolerite boulders is present (Smith et al., 2005), some of these boulders have been disaggregated and re-cemented into the conglomerate (Fig. 3C). Trapped and bound stromatolites are associated with strong terrestrial runoff.

[Fig.3]

4.3        Mineral Precipitated Stromatolites

Mineral precipitated stromatolites (which generally lack trapped and bound material) dominate the SPS Facies Association and (Table 2; Fig. 1). SPS grows on shore platforms, within chemically- or mechanically- produced pools and barrage pools (see: Forbes et al., 2010), constructed by stromatolite growth (Fig. 4), and the inclined apron terrace (Campbell et al., 2015) slopes that connect them (Fig. 4B). The water in the shore-platform pools varies from fresh to hypersaline (Smith et al., 2011; Perissinotto et al., 2014) depending on immediate wave and weather conditions. Elevated water temperatures, as much as 10º C above ambient environmental, at Cape Morgan, Luphatana and Richards Bay suggest inputs from warm thermal

spring activity (Smith et al., 2011). These warm waters often occur at the base of pools, separated from the overlying water by a thermocline. In such cases, SPS growth is absent below the thermocline (Smith & Uken, 2003).  At Cape Morgan and Tinley Manor, stromatolite colonies are clearly being fed by springs that emanate from a storm swash terrace, whereas at Luskentyre Bay, the colonies are fed from seeps flowing out of a peat bog (Fig. 5D). Water ponding, within suitable topography on shore platform surfaces provides accommodation space for growing SPS. SPS are variously present as thin crusts (1-30 cm thick), barrage deposits (Perissinotto et al., 2014), low mounds (>20cm high) and as oncoid-like cobble and boulder coatings (Edwards et al., 2017). These cobbles and boulders are probably the grinders involved in pothole production.

[Fig.4]

SPS growth zones are related to water physico-chemistry, calcification and limited to salinity values of <20 psu (generally 2– 10 psu) (Smith et al., 2011; Perissinotto et al., 2014; Edwards et al., 2017).  In areas where groundwater discharge is very strong coralline red algae may alternate with tufa growth in the lower part of cliffs reaching down into the lower intertidal zone. SPS may cease growing, desiccate and suffer rain dissolution, and then regrow forming erosion surfaces. Growth cessation may be due to "self" blocking of the water conduit by stromatolite growth and carbonate precipitation, wave erosion or drought conditions interrupting ground water inflow.

Three depth-controlled, stromatolite morphologies have been reported from pools (Fig. 4A) (Smith et al., 2011;Perissinotto et al., 2014;  Rishworth et al., 2016; Edwards et al., 2017). Partially emergent pustular stromatolites occur in the subareal wet area around pools and seeps (Fig. 4). This morphology occurs as mounds up to a few centimetres above pool rims and in very shallow water. At some localities (eg Luphatana and Mtentu: Fig. 1) the pustular stromatolite variety is not present.  In these cases water inflow is via joint planes directly into the pools. Pustular stromatolite morphology is often a link morphology between stromatolites and tufa (Fig. 4A). Laminar and columnar stromatolites (1-10cm high) occur in shallow water (Fig. 4A). These form a pool rim at, and just below, the water surface (Smith et al., 2011; Edwards et al., 2017). The laminar stromatolite morphology is particularly common in the wind-shadow margins of pools, whereas in deeper pools (20 – 30 cm depth) only the colloform stromatolite morphology is present (Fig. 4A).  SPS are highly colourful during bloom (Fig. 3), but become white on desiccation to a micrite crust.

The mineral precipitated SPS generally lack the particles that are present within the trapped and bound SPS (Fig. 5). The mineral precipitated SPS variation are characterised by a laminae comprising radiating cyanobacterial filaments, alternating with thinner concentric lamina (Fig. 5). Radial SPS lamination is on a centimetre scale, whereas the concentric lamination is much thinner (Fig. 5B). Mineral precipitated stromatolites may contain trapped and bound material but this is rare (Fig. 5D).

This suggests that the SPS trapped and bound and mineral precipitated stromatolites are end members of a continuous spectrum.

[Fog.5]

4.4     TUFA

Tufa waterfalls (see Perissinotto et al., 2014); varying from a few centimetres to several metres high, often coat cliffs and steep rock surfaces (Table 1; Fig. 3A & B). Tufa dominates on coast characterised by seacliffs (Table 2). At Cape Morgan an
inter layering of tufa and coralline red algae was noted in the lower part of the intertidal zone at a locality characterised by a strong water inflow.  If only cliffs are  present (and no shore platform) only tufa is present. Tufa waterfalls are often connected to SPS pools, either directly at the tufa toe or via connecting apron terraces.

# 5 GEOMORPHOLOGY & GEOLOGY OF SPS SETTINGS

All locations discussed are open coast except for Luskentyre Bay and Kuwait Bay (Table 2). The geomorphology of each shore platform is controlled by lithology, jointing and bedding style. Shore-platforms comprised a variety of competent lithologies (Table 2). Shore platforms vary from 5 to 60 m wide and are generally backed by a boulder beach or boulder ridge (Fig. 6). The boulder ridges contain angular blocks or megaclasts (up to 80 tons), as opposed to the smaller (> 50cm
diameter) rounded boulders found in boulder beaches and gullies in, and adjacent to, the shore platform (Table 2).

. The geomorphology of each shore platform is controlled by lithology, jointing and bedding style. Shore-platforms comprised a variety of competent lithologies (Table 2). Where the coast comprises incompetent lithologies no shore
platforms can form. Competent sandstones form wide shore-platforms (Fig. 6; Table 2). The SPS bearing shore-platforms at Mtentu and Luphatana (Fig. 6C; Table 2) are formed in well-indurated Lower Devonian Msikaba Formation sandstone. These are the widest at up to 60m wide. The Msikaba Formation is well-bedded (more-or-less horizontal) and vertically jointed. The shore-platform has been formed by wave quarrying of large blocks, which break along bedding and joint planes. The eroded boulders have accumulated in a boulder ridge at the rear of the platform (Fig. 6A).


[Fig.6]

At Cape Point SPS occurrences are rare but massive tufa deposits are present inland. Tufa is the main lithological component at Tofo and the shore-platform has been excavated into this lithology (Fig. 6B). Thin (1-2cm thick) sub-fossil stromatolites

(Table 2) are interbedded. Sub-fossil mineral precipitated stromatolites are also found as upper intertidal and supra tidal active pothole linings. No growing tufa or stromatolites were observed, but the presence of SPS in potholes within an active shore platform indicates them to be recent. It is possible that this tufa deposit is related to a lower sea level stillstand and has been reworked into a shore platform where SPS has developed at a later stage.

Dolerite sill shore Platforms are present at Tinley Manor, Ballito and Cape Morgan (Fig. 1); the latter (Fig. 7) being the best example (Table 2). The dolerite is strongly jointed and forms a rugged shore platform. The platform itself tends to undulate and shows minor sea cliffs and pools. The former are produced by wave bore quarrying of blocks along joint surfaces. Pools are formed by mechanical joint widening, pot-holing and chemical weathering while barrage pools are impounded by

stromatolite growth (Fig. 4).

At Port Edward (Fig. 1) granite forms a poorly developed shore platform littered with megaclasts. Minor SPS and tufa were noted near the landward shore-platform boundary. The Luskentyre Bay granitic, Harris Island, Scottish Hebrides (Fig. 1) shore platform is backed by an extensive but thin (± 30cm) peat bog. Seaward of this shore platform is a very extensive

intertidal fine-grained low-end macrotidal flat (Fig. 6).

At Richards Bay (Fig. 1) the southeast African coastline is marked by the semi-indurated Pleistocene Port Durnford Beds (Fig. 1). These comprise incompetent muds, silts and fine sands. The shore platform is poorly developed and backed by a retreating coastal cliff comprised of the poorly-consolidated Durban Beds sediments. Only tufa is present growing on the

cliff face but will not be preserved due to ongoing marine action.

**6. SPS Hinterland**

Storm swash terrace deposits (McKenna et al., 2012; Dixon et al., 2015) occur at the rear of Port Edward, Ballito, Tinley

Manor, Mtentu, Luphatana and Cape Morgan shore platforms. These are associated with boulder ridges, boulder beaches and coastal dunes. Storm swash terrace deposits (McKenna et al., 2012) may partially bury older beach ridges. Boulder beaches are present at the Luskentyre Bay, Harris, Scottish Hebrides (UK), SPS sites but here the hinterland is characterised by peat bog overlying bedrock which projects through at high points (Fig. 6). At Tofo (Mozambique) the tufa is backed by a very extensive (kilometres wide) coastal dune cordon, strongly impacted by farming and urbanization. Several SPS localities have

a coastal dune cordon hinterland and some are associated with bogs (Table 3). The hinterland at Giants Causeway comprises high basalt cliffs (Fig. 6). A model of the SPS Facies Association is shown in Fig. 7.

[Fig.7]

**7. Regional and Global Aspects**

SPS occur in a variety of climatic and oceanographic settings (Table 4) and although no systematic survey of SPS distribution has yet been undertaken, we expect that they are globally widespread. Shore platforms, associated with boulder ridges and boulder beaches, are indicative of high wave activity (Hall et al., 2006; McKenna et al., 2012; Dixon et al., 2015). In the case of the southern African eastern seaboard this is confirmed by modally high wave conditions (Guastella and Rossouw, 2012). Extreme waves, however, exceed modal conditions by several metres. On the southern African eastern seaboard, extreme waves originate from extra tropical low pressure systems, tropical storms/ cyclones (Smith et al., 2010). Tsunamis cannot be ruled out.

An eastern southern African seaboard high swell event on 18-20 March 2007 (impacts ranged from Port Elizabeth to Maputo) (Fig. 1), was produced by a cut-off-low pressure system. This event produced swells up to 14m high ($H_s$=8.5m) with run-ups of 7-11m amsl (Smith et al., 2007). A further high swell event between 31st August and the 1st September 2008 impacted the southern and southeast African coast, from Cape Point to Cape Morgan (Fig. 1). This was generated by an extremely deep low pressure system and produced swells of $H_s$= 10.7m (Guastella and Rossouw, 2012) with a 7-8m run-up at Cape Morgan (Smith et al., 2014).

Field inspection following both the March (2007) and September (2008) high swell events indicated changes to the Cape Morgan colonies (the other colonies listed in Table 1 were unknown prior to 2007). In the case of both storms, growing and unconsolidated microbial mat was largely removed by wave action, but lithified stromatolites remained on the shore-platform. However, large blocks of tufa were removed by the high swell impact. Following the March 2007 event, surface stromatolite growth at Cape Morgan had been largely restored by January 2008.

At Ballito (Fig. 1), sub-fossil peritidal stromatolites that had been buried under coastal reclamation were exposed by erosion during the 2007 high swell event. This event deposited a boulder beach (boulders >1m diameter) over the dolerite sill shore platform. At one point a thin crust of sub-fossil stromatolites was observed, this extended landward under the boulder beach and storm swash terrace. This shows that SPS can be interbedded with storm deposits. At Cape Morgan, boulders in the boulder beach are often coated by stromatolite growth indicating periods of disaggregation and stability. Several rounded boulders were found with multiple rims of stromatolite, indicating they had been experienced several cycles of stability and movement.

At Luphatana extant SPS colony is located just in front of a boulder ridge which contains 80 ton boulders, indicative of extreme waves (date unknown), the parameters of which are as yet unquantified (Fig. 5A). Similarly the St Johns Point and

Giant's Causeway, N. Ireland stromatolites (Cooper et al., 2013) are both associated with very large boulders. Wave spray deposits have been found at levels of 60m amsl at Cape Morgan, South Africa (Smith et al., 2014) and at Aird Uig (Harris, Scotland) wave scouring has taken place on a cliff top at a height of 20–30 m OD (Hall et al., 2006), proving that SPS can withstand the present extremes in wave climate.

The Irish and Scottish coasts are known for strong wave action (Hall et al., 2006; O'Brien et al., 2012) as the 2013-14 storm season has proved (Wadley et al., 2015). Both Giant's Causeway (on the west coast) and Luskentyre Bay are afforded some protection, but the boulder storm beaches at both localities (Table 2) show then to be vulnerable to wave attack.

It must be restated that this sample is far from complete, representing only coastlines that have been investigated but from Table 4 the following can be noted. Extant SPS are:
  ➢ Recorded in passive plate margins and epeirogenic settings
  ➢ Occur from microtidal to macrotidal environments
  ➢ Sea level rise varying from 0.07 to 2.89 mm/yr
  ➢ Sea surface temperatures varying from 7.4 to 32.3º
  ➢ Cool Temperate to Dry Desert.
  ➢ Varying high wave regimes, all chracterised by boulders

# 8 Discussion

 SPS grow in shore platform depressions on high-energy coasts.  They grow within shallow rock pools (Smith et al., 2011; Perissinotto et al., 2014; Rishworth et al., 2016) potholes and SPS dammed barrage pools (Forbes et al., 2010; Perissinotto et al., 2014) on shore platforms. SPS comprise thin micritic crusts, with only rare examples of the trapped and bound stromatolite varieties being associated. Not all localities show the complete tripartite stromatolite morphology (Fig. 4A). The pustular variety may be absent but the subaqueous laminar and columnar and the colloform variety are ubiquitous. Where vertical surfaces are present tufa grows.

Any SPS model must take into account that they are forming at the interface of freshwater seeps and a high energy rocky peritidal zone (Smith et al., 2011; Perissinotto et al., 2014). The SPS are calcium carbonate mineralised due to the high pH regime (Smith et al., 2011) and are growing on older siliceous rocks (Tofo may be an exception but the base is not visible) within a siliciclastic contemporary setting. Mud and sand are passing through this system but are not being deposited in

significant amounts. Microbialites are ubiquitous in the peritidal setting but only become stromatolites if a carbonate rich groundwater plume is present (Smith et al., 2011; Rishworth et al., 2016).

The following points also need to be considered, although they may, or may not, be necessary for SPS growth. Sea level rise
is ongoing. SPS possess seasonal to sub-seasonal (due to storm abrasion) laminae (Smith et al., 2005). The lack of climax lamination (Reid et al, 2000), characterised by diatoms (Smith et al, 2005) in the calcified form may be due to storm activity or non-calcification of this lamina type. Warm thermal ground water may be present indicating a groundwater source. The stromatolites and shore-platform contact is an unconformity.

The diversity of shore platform substrates suggests that lithology is not important for SPS growth; however, the competency of the substrate is vital for growth and probably for potential preservation. Tufa deposits and pustular stromatolite deposits are very unlikely to survive transgression or regression as there is no accommodation space. However, SPS in rock pools and barrage pool build ups can form a lenses or layers of stromatolite which could then be overstepped during marine tansgression (possibly similar to the multi-metre steps which post-dated the last glacial maximum) and
preserved in a future stratigraphy. Marine processes may break up the stromatolite and free stromatolite encrusted boulders from barrage and rock pools, such as is seen at Cape Morgan (Fig. 2A), and transport them as littoral drift for deposition elsewhere as conglomerates. SLR rise is taking place globally, thus SPS deposits could form as part of a global transgressive coastal sequence.

The best opportunity for preservation is provided by rock pools, especially potholes, in competent shore-platform rock as this provides accommodation space. SPS growth itself could seal the SPS deposit, especially in flat bedded competent sandstone as at Mtentu and Luphatana (Fig 5). Thus SPS environments could be preserved as lenses on palaeo-shore-platforms. The extant stromatolite regrowth which has taken place at Ballito (Fig. 1) suggests that preservation can take place. An investigation of subaqueous post-last glacial maximum coastlines might resolve this issue.

## 8.1 Global Extant Stromatolites

There are several important differences between the extant SPS described here and other, well known extant stromatolite occurrences. Variations on the Hamelin Pool theme are generally used for artists' conceptions of the Archean; however this
instance is unusual as within a hypersaline lagoon (Logan et al., 1964). The Highborne Cay, Bahamas sub-tidal model is characterized by columnar stromatolites within an ooid shoal (Reid et al., 1995; Visscher et al., 1998; Baumgartner et al., 2009). Both Shark Bay and the Bahamian extant stromatolite settings contain stromatolites produced by trapping and binding within a soft coastline. These may leave traces such as MISS (Noffke and Awramik, 2013) but are unlikely to be preserved

in the long-term. Kremer et al (2008) found calcium carbonate being precipitated within annual Cyanobacteria mats in beaches of the Dutch North Sea, but again these were of the trapped and bound variety, as opposed to mineral precipitation. The Kuwait Bay examples (Alshuaibi et al., 2015), however, are of the mineral precipitation variety. These are developed on beachrock and could conceivably be preserved.

## 8.2 Precambrian Stromatolites

Precambrian stromatolites are commonly found to have developed in transgressive settings on varying substrate types (Table 5). Erosion surfaces are common within them (Kranendonke (2011; Nutman et al., 2016) as is the case with SPS. In
the case of Strelley Pool (Australia), stromatolites formed initially on a rocky shore-platform (Allwood et al., 2006), which compares well with SPS (Tables 1 & 2). Allwood et al. (2006) base this interpretation on the following:

1. Wide and discontinuous distribution of boulders with a rounded clast-supported fabric,
2. Correlation between clast and substrate, on an unconformity,
3. Substrate-dependent lateral transition from clustered and isolated large boulders on a shore platform to embayment beach conglomerates type
4. The presence of palaeo-cliffs, fissures and cavities in the substrate
5. Soft mud intraclasts and desiccation cracks associated with local mudstone substrate

The extant SPS facies association demonstrates all the points made by Allwood et al. (2006) concerning the Strelley Pool stromatolites (Tables 1 & 2) and further the stromatolite morphological scale of the two is similar. Although the SPS horizontal exposures vary from 10s to 100s of metres, they are time equivalents and, if conditions where right, could develop into beds stretching for 10s or 100s of kilometers. The presence of "stromatolites" on vertical surfaces at Strelley Pool is interesting as this suggests that these stromatolites were associated with tufa. This shows that tufa can be preserved despite
the perceived lack of accommodation space. The 1.88 Ga Gunflint Chert is based on a weathered Archean lava, characterized by rounded lava boulders (Brasier et al., 2015) and may also be a palaeo SPS occurrence.

In contrast to SPS most Precambrian stromatolites apparently formed on soft coastlines (Table 5). Perhaps the Precambrian
marine climate was markedly less aggressive than at present or stromatolites are simply preserved in low energy embayments such as Kuwait Bay (Alshuaibi et al., 2015) or Luskentyre Bay.

The presence of tufa within the SPS facies association and its similarity to that of the 3.4 Ga Strelley Pool stromatolites strongly hints that microbial life existed in Archean terrestrial settings. It is quite reasonable to assume that tufa was present landward of Archean stromatolite settings, but was not preserved.

## 8.3 SPS AS A MODERN PRECAMBRIAN ANALOGUE

It has been suggested that no adequate marine extant stromatolite analogue exists for Precambrian marine mineral precipitated stromatolites, as most modern marine extant stromatolites are of the trapped and bound variety (Awramik & Grey, 2005). The mineral precipitated SPS deposits fill this gap. Most Precambrian stromatolites formed in a transgressive setting, similar to the SPS (Table 5). The peritidal stromatolite setting is a common theme for Precambrian stromatolites; although intertidal and subtidal components are also present (Table 5). The SPS setting shows only a stunted intertidal and no subtidal component. Microbialite develops in the crack between terrestrial and marine influence where Metazoans are reduced by the extreme nature of this setting and thus reduces the competition for space. If the ground water chemistry is suitable, SPS form. It is easy to imagine how the SPS environment would unfold if Metazoan activity was reduced or not present, as was the case in the Precambrian; this would allow large subtidal stromatolites, such as those in the Malmani Dolomite (Sumner & Grotzinger, 2004) (Table 5) to develop. Similarly it is very likely that tufa existed in the terrestrial environment, but has not been preserved (or recognised). The SPS model strongly suggests the occurrence of terrestrial prokaryotes during the Archean. Extant SPS are growing during a mild transgression, thus their preservability at the present time is probably low. However, should SLR accelerate in the future, it is possible that this environment could be overstepped and preserved as rock pool and pothole fills. The lower colloform and laminar stromatolite variety are more likely to be preserved and the tufa least likely or not at all.

## 8.4 Extraterrestrial Implications

Phosphorous on Mars is commoner than on Earth (Greenwood et al., 2007) so it is possible that simple life could have been present. If this was the case then stromatolites could have formed if life (as we know it) was ever present, and the water chemistry was suitable. On Mars, marine and lacustrine flooding surfaces should be investigated. Palaeoshore-platforms, especially if associated with megaclast fields or ridges should be targets for investigation. Although no shore platforms have been reported (Banfield et al., 2015) as yet, the terrain at Chryse Planitia and Arabia Terra, which bordered the postulated Vastitas Borealis Ocean, (Wilson et al., 2016, Rodriguez at al, 2016), could be shore platform candidates. Recently Ruff & Farmer (2016) suggested that structures in the Ma'Adim terraces within the Gusev Crater are siliceous stromatolites. These terraces may have been formed by wave action in the Gusev Palaeolake or at the margin of the Vastitas Borealis Ocean and may be an extraterrestrial palaeo-SPS environment.

# 9 Conclusions

Extant SPS are present on high energy rocky coasts on passive margins and in epeirogenic settings; however this may be a function of sampling. They are associated with well-indurated shore platforms, boulder bars and boulder beaches. The hinterland is frequently boggy or marshy. SPS appear to be unrelated to climate and tidal regime. They may be better developed in warmer climates but the sample size is yet too small. If vertical surfaces are present on the shore platform, tufa forms. In shallow shore platform pools SPS develop. There is a gradation from SPS to tufa. SPS develop within transgressive settings, as was the case with many Precambrian stromatolites. The preservability of these micrite stromatolites is probably low, but may improve with stromatolite cementation to competent substrate rock. Contemporary rocky and Quaternary coastlines should be investigated globally for the SPS environment. The SPS setting is a valid analogue for the Archean Strelley Pool Archean stromatolite and possibly the base of the Gunflint stromatolite occurrence, however most Precambrian stromatolites appear to have formed on soft coastlines. The association with trapped and bound stromatolites and the Luskentyre Bay environment (shore platform and tidal flat) hints at a possible SPS soft coastline link. Shore-platform settings should be targets in the search for Martian stromatolites.

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

## Tables

Table 1: Location, context and stromatolite type of SPS localities (Y=yes; N= no; n/r = no record).

| LOCATION | COORDS | SHORE PLATFORM | CLIFF | TRAPPED & BOUND | MINERAL PRECIPITATION | TUFA |
|---|---|---|---|---|---|---|
| Cape Point, SA | 34º 20' 48.42"S; 18º 27' 48.16"E | Y | Y | | | Y |
| [1]Storms River | 34º 10' 34" S; 24º 39' 46" E | Y | n/r | N | Y | n/r |
| [1]Oyster Bay | 34º 11' 25.07" S; 24º 41' 43.76" E | Y | | N | Y | n/r |
| [1]Cape St Francis | 34º 12' 49" S; 24º 50' 04" E | Y | N | N | Y | n/r |
| [1]Seaview | 34º_01' 03.16" S; 25º_21' 56.48" E | Y | N | N | Y | n/r |
| [1]Skoenmakerskop | 34º_02' 28.23" S; 25º_32' 18.60" E | Y | N | Y | Y | n/r |
| [1]Cape Recife | 34º_02' 42.13" S; 25º_34' 07.50" E | Y | N | N | Y | n/r |
| [2]Cape Morgan | 32º 41' 36" S ; 28º 22' 27" E | Y | Minor | Y | Y | Y |
| Luphatana | 31º 25' 10" S; 29º 51' 30" E | Y | N | N | Y | N |
| Mtentu | 31º 14' 30" S ; 31º 03' 22" E | Y | N | N | Y | N |
| Port Edward | 31º 02' 55.26" S; 30º 13 '47.79" E | Y | N | N | Y | N |
| Ballito | 29º 32' 15" S; 31º 13' 20" E | Y | Minor | N | Y | N |
| Tinley Manor | 29º 26' 43.98"S; 31º 17' 26.99"E | Y | Minor | N | Y | Minor |
| Richards Bay | 28º 46' 15.20"S 32º 08' 00.32"E | X | Y | N | Y | Y |
| Tofo, Mozambique | 23º 41' 26" S; 35º 33' 04" E | Y | Y | N | Y | N |
| [3]Giant's Causeway, N. Ireland, UK | See reference | Y | Y | N | Y | N |
| St John's Point, N. Ireland, UK | 54º 13' 31.23"N; 5º 39' 32.69"W | Y | Y | N | Y | N |
| Luskentyre Bay (S), Harris, UK | 57º52'08.73"N 6º57'42.01"W | Y | Y | N | Y | Y |
| Luskentyre Bay (N), Harris, UK | 57º52'17.65"N 6º54'28.52"W | Y | N | Y | Y | N |
| Northton, Harris, UK | 57º48'09.56"N 7º04'57.52"W | N | Y | N | Y | Y |
| [4]Kuwait Bay, Kuwait | See reference | Y | N | N | Y | N |
| [5]N. Sea, Netherlands | See reference | N | N | n/r | ? | N |
| [6]SW Australia | See reference | Y | Y | N | Y | Y |
| Monkey Mia, W. Australia | Not known | Y | N | N | Y | N |

[1] Perissinotto et al. (2014)
[2] Smith & Uken (2003)
[3] Cooper et al. (2013)
[4] Alshuaibi et al., (2015)
[5] Kremer et al. (2008)
[6] Forbes et al. (2010)

Table 2: Lithological features of Shore Platforms (N/R: no record)

| LOCATION | LITHOLOGY | ATTITUDE | BOULDERS PRESENT | AGE | BEDROCK STATUS |
|---|---|---|---|---|---|
| Cape Point, SA | Sandstone (Nardou) | Folded | Boulder beach | Ordovician | v.indurated |
| Storms River | Sandstone (Nardou) | Folded | Boulder Ridge | Ordovician | v.indurated |
| Oyster Bay | S/Stone (Nardou) | n/r | n/r | Ordovician | v.indurated |
| St Francis | S/Stone (Nardou) | | n/r | Ordovician | v.indurated |
| Seaview | [1]SQuartzites, Grits and Phyllites | Deformed | Boulders in pools | Precambrian | v.indurated |
| Skoenmakerskop | [1]Metaseds (Sardinia Bay Fm) | Deformed | n/r | Ordovician | v.indurated |
| Cape Recife | S/Stone (Nardou) | Deformed | n/r | Ordovician | v.indurated |
| [2]Cape Morgan | Dolerite | Sill | Boulder Ridge | Jurassic | Fresh |
| Luphatana | S/Stone (Msikaba) | Horizontally bedded | Boulder Ridge | L Devonian | v. ndurated |
| Mtentu | S/Stone (Msikaba) | Horizontally bedded | Boulder Ridge | L Devonian | v.indurated |
| Port Edward | Granite | Deformed | Boulder Ridge | 1.1 Ga | Fresh |
| Ballito | Dolerite | Sill | Storm Beach | Jurassic | Fresh |
| Tinley Manor | Dolerite | Sill | Storm Beach | Jurassic | Fresh |
| Richards Bay | Pt Durnford Fm muds, silts, F/sst | Sea cliff | Storm Beach | Pleistocene | Semi-consolidated |
| Tofo, Mozambique | Tufa (no base seen) | Massive | Storm Beach | Pleistocene | Tufa |
| [3]Giants Causeway, N. Ireland | Basalt | Columnar Basalt | Storm Beach | Cretaceous | Fresh |
| St John's Point, N. Ireland | Limestone | n/r | Storm Beach | Lower Carboniferous | Fresh |
| Luskentyre Bay (North), Harris Island, UK | Granitic | Deformed | Storm Beach | Proterozoic-Archean | Fresh |
| Luskentyre Bay (South), Harris Island, UK | Granitic | Deformed | Storm Beach | Proterozoic-Archean | Fresh |
| Northton, Harris Island, UK | Granitic | Sea cliff | N | Archean | Fresh |
| [4]Kuwait Bay, Kuwait | Beach Rock | Bedded | Not present | Late Quaternary | Fresh |
| [5]SW.Australia | Limestone/ Granite | n/r | Storm Beach | Pleistocene Limestone/ Granite (540-780 Ma) | Fresh |
| Monkey Mia, W. Australia | Not known | n/r | Not present | n/r | Fresh |

[1] Perissinotto et al. (2014)
[2] Smith & Uken (2003)
[3] Cooper et al. (2013)
[4] Alshuaibi et al., (2015)
[5] Forbes et al. (2010)

Table 3: Hinterland geology & geomorphology of SPS locations (gaps indicate not recorded)

| LOCATION | SHORE PLATFORM | CLIFF | STORM SWASH TERRACE | DUNES | PEAT BOG | TUFA | URBANISED |
|---|---|---|---|---|---|---|---|
| Cape Point, SA | Y | Y | N | N | n/r | Y | N |
| [1]Storms River | Y | n/r | n/r | n/r | n/r | | N |
| [1]Oyster Bay | Y | N | n/r | Y | n/r | | Partly |
| [1]Cape St Francis | Y | N | n/r | Y | n/r | | Y |
| [1]Seaview | Y | N | n/r | Y | n/r | | Y |
| [1]Skoenmakerskop | Y | N | n/r | Y | n/r | | N |
| [1]Cape Recife | Y | N | n/r | Y | n/r | | N |
| [2]Cape Morgan | Y | Y | Y | Y | Y | | N |
| Luphatana | Y | N | Y | Y | Y | | N |
| Mtentu | Y | N | Y | Y | Y | | N |
| Port Edward | Y | N | Y | Y | Y | | N |
| Ballito | Y | Minor | Y | N | Y | | Y |
| Tinley Manor | Y | Minor | Y | N | Y | | Y |
| Richards Bay | N | Y | N | Y | N | | N |
| Tofo, Mozambique | Y | Y | N | Y | N | | Y |
| [3]Giant's Causeway, N. Ireland | Y | Y | Y | N | N | | Some infrastructure |
| St John's Point, N. Ireland | Y | Y | n/r | n/r | N | | N |
| Luskentyre Bay (S) Harris, UK | Y | Y | N | N | Y | | N |
| Luskentyre Bay (N) Harris, UK | Y | N | N | N | Y | | N |
| Northton Harris, UK | N | Y | N | N | Y | | N |
| [4]Kuwait Bay, Kuwait | Y | N | n/r | Y | N | | N |
| [5]N. Sea, Netherlands | N | N | n/r | n/r | N | | N |
| [6]SW Australia | Y | Y | n/r | n/r | n/r | | N |
| Monkey Mia, W. Australia | Y | N | n/r | Y | N | | N |

[1] Perissinotto et al. (2014)
[2] Smith & Uken (2003)
[3] Cooper et al. (2013)
[4] Alshuaibi et al., (2015)
[5] Kremer et al. (2008)
[6] Forbes et al. (2010)

Table 4: Tectonic and Oceanographic Aspects of SPS localities. Tidal data from the South African Naval Hydrographic Office and British Admiralty (*measured values; [#]Sea level rise and [##]Sea surface temperature) and satides.co.za (modeled data)(n/g=not gauged).

| LOCATION | TECTONIC | [#]SLR (mm/yr) | [##]SST RANGE (°C) | TIDAL RANGE (m) | WAVE REGIME | CLIMATE ZONE |
|---|---|---|---|---|---|---|
| Cape Point | Passive | 1.94 (Simon's Town) | 8 – 15 | 1.83* | Very High | Mediterranean |
| [1]Storms River | Passive | n/g | 15.5 - 19.5 | 2.14 | Very High | Moderate coast |
| [1]Oyster Bay | Passive | n/g | 16 - 18 | 2.12 | Very High | Moderate coast |
| [1]Cape St Francis | Passive | n/g | 16 – 20.5 | 2.1 | Very High | Moderate coast |
| [1]Seaview | Passive | n/g | 16 - 20 | n/g | Very High | Moderate coast |
| [1]Skoenmakerskop | Passive | n/g | 16 - 23 | n/g | Very High | Moderate coast |
| [1]Cape Recife | Passive | 2.39 (PE) | 16 - 23 | 1.99 * | Very High | Subtropical coast |
| [2]Cape Morgan | Passive | n/g | 17.5 - 20.5 | 1.98 | Very High | Subtropical coast |
| Luphatana | Passive | n/g | 19 – 22.5 | n/g | Very High | Subtropical coast |
| Mtentu | Passive | n/g | 19 – 22.5 | n/g | Very High | Subtropical coast |
| Port Edward | Passive | n/g | 19.5 - 23 | 2.05 | Very High | Subtropical coast |
| Ballito | Passive | 1.23 (Durban) | 20.5 - 24.5 | 2.13* | Very High | Subtropical coast |
| Tinley Manor | Passive | n/g | 20.5 - 24.5 | n/g | Very High | Subtropical coast |
| Richards Bay | Passive | n/g | 20 – 24.5 | 2.3 | Very High | Subtropical coast |
| Tofo | Passive | n/g | 22 - 29 | 5.0 | Very High | Tropical coast |
| [3]Giant's Causeway | Epeirogenic | 0.07 (Dublin) | 7.9 – 15.3 | 2.4* (Port Rush) | High | Cool Temperate |
| St John's Point | Epeirogenic | 0.07 (Dublin) | 7.4 – 15.4 | 2.4* (Port Rush) | High | Cool Temperate |
| Luskentyre Bay (S) | Epeirogenic | 1.92 (Stornoway) | | 4.4* (Tarbert) | Moderate | Cool Temperate |
| Luskentyre Bay (N) | Epeirogenic | 1.92 (Stornoway) | | 4.4* (Tarbert) | Moderate | Cool Temperate |
| Northton | Epeirogenic | 1.92 (Stornoway) | | 4.4* (Tarbert) | Very High | Cool Temperate |
| [4]Kuwait | Epeirogenic | ? | 13.3 - 32.3 | 4.3* (Kuwait Bay) | Low | Dry Desert |
| [5]SW Australia | Passive | 1.54 (Freemantle) | | ? | ? | Mediterranean |
| Monkey Mia, W. Australia | Epeirogenic | 2.89 (Carnarvon) | | ? | ? | Dry Desert |
| [6]N. Sea, Netherlands | Epeirogenic | 1.71-2.53 (Netherlands) | | ? | ? | Cool Temperate |

[1] Perissinotto et al. (2014)
[2] Smith & Uken (2003)
[3] Cooper et al. (2013)
[4] Alshuaibi et al., (2015)
[5] Forbes et al. (2010)
[6] Kremer et al. (2008)
Sea Surface Temperature data obtained from Smit et al. (2013) and Alshuaibi et al. (2015).

Table 5: Comparison of SPS with some Precambrian stromatolites.

| ROCK UNIT | AGE (Ga) | ENERGY | PALAEO-TIDAL REGIME | COAST | PALAEO-SUBSTRATE | TRANSGRESSION | SOURCE |
|---|---|---|---|---|---|---|---|
| Isua Group Greenland | 3.7 | ? | Shallow marine | Soft? | Metacarbonate | ? | Nutman et al. (2016) |
| Strelley Pool Chert, N. Aus | 3.43 | Hi | Peritidal | Rocky (SPS) | Jasper banded black chert; potholes | Transgression | Allwood et al., (2006) |
| Josefsdal Chert, SA | 3.5-3.3 | Hi | Peritidal | Soft? | Volcanic seds / Hydrothermal spring source | ? | Westall et al (2006) |
| Moodies Group, SA | 3.22 | Med | Inter to subtidal | Soft (gravel) | Sand | Transgression | Gamper et al (2011) Ericksson et al (2006) |
| Pongola Group, SA | 2.9 | Low | Inter-to-subtidal | Soft? | Sand | Transgression? | Mason & Von Bruun (1979) Bolhar et al. (2015) |
| Malmani Dolomite, SA | 2.56-2.52 | Low-Hi | Peritidal to deep subtidal- | Soft? | Variable | Transgression? | Sumner & Grotzinger (2004) |
| Turee Creek Group, W. Australia? | 2.45–2.22 Ga | Hi | Sub-tidal | Soft? | Sandy siliciclastic/ carbonate | Progradation | Martindale et al (2015) |
| Gunflint chert, Canada | 1.88 | | | | Archean lava with rounded boulders | Transgression | Brasier et al. (2015) |
| Extant SPS | Extant | Very Hi | Peritidal | Rocky (SPS) & T& B v. rare) | Dolerite, Sandstone, Tufa Granite Sand & Gravel (v. rare) | Transgression | Smith & Uken (2003) Perisotimo et al. (2014) This study |

**Figure Captions**

Fig. 1: Location of SPS sites on the eastern seaboard of southern Africa, Northern Ireland and the Scottish Hebrides.

Fig. 2: A: Shore platform showing a low tufa curtain (T), barrage (B) and pool (BP); B: Higher tufa curtain (T) forming on a cliff at the back of the dolerite shore platform with stromatolites (S) at its toe (A and B from Cape Morgan), and C: Tufa curtain from Northston, Harris, UK. Here there is no shore platform.

Fig. 3:  A: Trapped and bound stromatolites forming on a stromatolite apron, located on the Luskentyre Bay shore platform, Harris, UK. The stromatolite is growing around cobbles swept onto the apron by inflowing water; B: trapped and bound stromatolites developed on a cobble beach adjacent to a shore platform. Note the shell debris which has been bound into the Cape Morgan stromatolite and, C: Dolerite boulders quarried from the  shore platforms and bound by stromatolite growth. Fig. B & C are reproduced from Smith et al., 2005; 2011).

Fig. 4: SPS mineral precipitated stromatolites. ; A: Stromatolite pool showing pustular (1); laminar and columnar (2) and colloform (3) stromatolite types from Cape Morgan, South Africa; B: stromatolite pool (SP), stromatolite apron (A) and stromatolite rim (R);  C: joint controlled stromatolite pool (SP) and stromatolites and a stromatolite rim  ( R). Both B and C are from Luskentyre Bay, Harris, UK and, D: Stromatolite pool from Mtentu, South Africa.

Fig. 5: A: Thin-section image of  a mineral precipitated stromatolite (35 X magnification); B: This image shows a domical calcified stromatolite (scale bar is 1mm); C: Image of growing microbialite cyanobacteria  filaments (scale bar is 10μ) and D: Example of rare trapped and bound grains within microbialite which may have been introduced by wave or wind action (scale bar is 400μ).


Fig. 6: A: Sandstone shore platform and boulder ridge at Luphatana, South Africa. Stromatolites are growing in the pool (arrowed). Eighty ton boulders are present in the boulder ridge showing the contrasting shore platform environmental extremes. B: Tofo, Mozambique tufa shore platform with scattered large boulders. C: Giant's Causeway basalt shore platform and cliffs. D: Lewisian gneiss shore platform at Luskentyre Bay, Harris, UK. Note the peat marsh (P).


Figure 7: SPS Facies Association model based around the Cape Morgan SPS context.