# Peer review of "The Extant Shore-Platform Stromatolite (SPS) Facies Association: A glimpse into the Archean?"

_Biogeosciences, 2017_

## Referee Comment (RC1) · Anonymous Referee #1 · 8 Aug 2017

The manuscript "Extant shore-platform stromatolite (SPS) assemble" by Alan Smith et al. observed SPS lithological and geomorphological assemblage and described that SPS are produced by mineral precipitation. However, the data provided by the authors are not sufficient to support interpretations and conclusions, at least in the way they are described in the manuscript in the present form. The present data are definitively publishable but not to a well-known journal as Biogeosciences.

General commentsïijŽ 1. The microorganisms were prevalent in the Precambrian. Microbial fossils present the cellular structure, which is similar to cyanobacteria and other prokaryotes. But morphological analysis of these microbial fossils is often not enough to obtain the correct information. The acquisition of information would highly benefit from in-situ analytical techniques such as laser Raman spectroscopy, and stable

isotopes. The potential effect of microbes in the formation of stromatolites should be further discussed. 2. The scientific methods and assumptions are not valid and clearly outlined, such as when the authors try to discuss SPS preservation potential in the geological record, what is the preservation mechanism? 3. The formation of stromatolites requires certain environmental conditions, especially more information about the water chemistry measurements should be provided. Specific comments: 1. Typing errors p.8, l. 33 ("be necessary for SPS growth" miss symbol ) and p18, l10 "and Precambrian Examples.?" 2. The manuscript has also a number of grammatical errors that need to be corrected.

---

## Referee Comment (RC2) · Anonymous Referee #2 · 19 Aug 2017

In this manuscript the authors compare different stromatolite assemblages along the south-east African shoreline, compare in situ factors under which they form, and evaluate their potential to gain further understanding of Precambrian stromatolite formation as well as their potential as an indicator of previous life on Mars. While the topic in general is interesting and a comparison of recent and ancient stromatolite formation across different geological settings may make an important contribution to our knowledge of this field, the paper appears rather descriptive and lengthy and may benefit from some restructuring and focusing. In the first place, I would suggest to modify the title in a way that it at least contains more specific information about this study or reflects the major outcome. Similarly, the abstract appears as a listing of findings of this study. Here, a clear statement of the motivation of this study and highlighting the major outcome in

one concluding sentence should be added. The motivation of this work is stated rather clearly at the end of the introduction section. However, especially in the results section, a stronger structuring along the original research objectives would help the reader find their way through this large amount of detailed site information which is provided in the results section. Although some of the site-related information is already presented in tables, comparison of key features across sites would benefit from a more condensed presentation in tables rather than in text. This way it would be easier for the reader to recognize in which key aspects the different study sites differ and what might be the most important regulating factors for stromatolite formation. This would also help the authors to check carefully if really all the aspects that they provide in the results section are needed for the discussion. The discussion is already written quite concisely, however, a more direct reference to the objectives stated in the introduction would be helpful. For example, the discussion of the potential of SPS as Precambrian analogues remains rather superficial. In addition, I have two concerns regarding the integration of aspects of microbiology. The authors integrated a longer section about the role of prokaryotes in stromatolite formation and the importance of the competition between prokaryotes and metazoans. However, this aspect is not targeted at all in the results section and only occasionally addressed in the discussion, and I was wondering why it was introduced so thoroughly in the introduction. In addition, some of the statements in this paragraph of the introduction (p. 2, l. 5-14) are not correct or are not sufficiently explained. What is meant by the statement that "Prokaryotes, however, do not react well to Metazoan competition"? (p. 2, l. 6). In line 10-11 "..but under contemporary conditions they can only thrive in extreme environments that limit Metazoan competition". Given the fact that you find about 10_10 bacteria per gram forest or grassland soil, this statement does not hold or its meaning in this context here should be clarified.

Specific comments: p. 2, l. 4: "more plausible Precambrian stromatolite analog" - more plausible compared to what? p. 3, l. 25-27: What does it mean that prokaryotes dominate? What about unicellular eukaryotes in such environments? p. 4, l. 9: Which coastline, please specify. p. 4, l. 26-35: It is not clear which sites this information refers

to. p. 8, l. 31-32: This is a rather vague statement: Why should these factors then be considered?

---

## Author Comment (AC1) · 22 Aug 2017

Referee 1 made three interesting comments

1. Data on the micro-organisms could be improved. This study was secondary to the major SPS study. However a Raman SEM process is currently underway.

2. The methods and assumptions not clearly outlined. Ref 1 singled out the preservation method. To be frank we do not know. All we can say at this point is that transgressive sequences can be preserved. The exact mechanics? We will definitely reconsider this.

3. More water chemistry data required. This was not the focus of the study it was to describe a lesser known modern enviro/ geomorphological setting of SPS. Data on this

field has been published in Smith et al (2011) and Rishworth et al (2016 & 2017). This research is ongoing.

4. We will certainly revisit the Precambrian similarities and the Martian kites!

Thank you for your participation and I apologise for the length of time it took me to reply

---

## Author Comment (AC2) · 9 Oct 2017

In this manuscript the authors compare different stromatolite assemblages along the south-east African shoreline, compare in situ factors under which they form, and evaluate their potential to gain further understanding of Precambrian stromatolite formation as well as their potential as an indicator of previous life on Mars.

While the topic in general is interesting and a comparison of recent and ancient stromatolite formation across different geological settings may make an important contribution to our knowledge of this field, the paper appears rather descriptive and lengthy and may

benefit from some restructuring and focusing.

Reply: I accept the paper needs focusing and shortening. Because this paper is trying to distill a facies association from a relatively newly documented Marine mineral precipitated stromatolite association it has to be descriptive. SPS is the only growing marine stromatolite environment that can be directly compared to an Archean marine fossil palaeoenvironment, ie Strelley Pool (Allwood et al. (2006) who interepreted the 3.4Ga stromatolite as stromatolites developed on a shhore platform (wave-cut platform). This makes the SPS setting very important.

The SPS are mineral precipitated stromatolites, whereas Shark Bay and the Carribean are of the trapping and binding variety. Thus the SPS setting lends itself to a direct comparison with Archean stromatolite occurrences.

Work has been done on Cape Morgan, South Africa (Smith & Uken, 2003; Smith et al, 2005; 2011) and the Giants Causeway SPS (Cooper et al., 2013) and other SPS from the Eastern Cape, South Africa (Perissinotto et al., 2014; Rishworth et al., 2016; 2017; Edwards et al 2017) but no attempt has been made for a detailed comparison of all the known SPS occurrences. We do this in this paper hence it is very descriptive.

This paper compares all known SPS occurrences to determine similarities and differences. From this we try to distill an SPS facies association. As this is work on a new marine stromatolite setting it has to lean heavily on fieldwork and facies analysis and is thus descriptive.]

Ref 2: In the first place, I would suggest to modify the title in a way that it at least contains more specific information about this study or reflects the major outcome.

Reply: I agree to this. Perhaps the following might be better:

Geomorphological and Stratigraphical Aspects of the Extant marine shore-platform stromatolite (SPS) assemblage and comparison with certain Archean examples.

Ref 2: Similarly, the abstract appears as a listing of findings of this study. Here, a

clear statement of the motivation of this study and highlighting the major outcome in C1 BGD Interactive comment Printer-friendly version Discussion paper one concluding sentence should be added.

Reply: I agree with this and it can be done.

Ref 2: The motivation of this work is stated rather clearly at the end of the introduction section. However, especially in the results section, a stronger structuring along the original research objectives would help the reader find their way through this large amount of detailed site information which is provided in the results section.

Reply: I agree with this and it can be done.

Ref 2: Although some of the site-related information is already presented in tables, comparison of key features across sites would benefit from a more condensed presentation in tables rather than in text. This way it would be easier for the reader to recognize in which key aspects the different study sites differ and what might be the most important regulating factors for stromatolite formation. This would also help the authors to check carefully if really all the aspects that they provide in the results section are needed for the discussion.

Reply: I think that this is a very good idea and should be explored. Perhaps the large table can be split up into several smaller with only the more important tables displayed in the main text, others can be in a supplement.

Ref 2: The discussion is already written quite concisely, however, a more direct reference to the objectives stated in the introduction would be helpful. For example, the discussion of the potential of SPS as Precambrian analogues remains rather superficial.

Reply: This can certainly be beefed up.

Ref 2: In addition, I have two concerns regarding the integration of aspects of microbiology. The authors integrated a longer section about the role of prokaryotes in

stromatolite formation and the importance of the competition between prokaryotes and metazoans. However, this aspect is not targeted at all in the results section and only occasionally addressed in the discussion, and I was wondering why it was introduced so thoroughly in the introduction.

Reply: I accept this criticism and this section can be reduced. The purpose of the paper is not to prove that these are stromatolites as this has already been done and the relevant papers are quoted. The purpose of this paper is to fully describe the SPS geomorphological setting and to indicate its global (Archean and possibly extraterrestrial) significance. I can see that the introduction lacks a strong description on the geomorphological and stratigraphical focus of this paper. This needs to be clearly stated at the outset as the referees are clearly expecting a strong biological focus to follow.

Ref 2: In addition, some of the statements in this paragraph of the introduction (p. 2, l. 5-14) are not correct or are not sufficiently explained. What is meant by the statement that "Prokaryotes, however, do not react well to Metazoan competition"? (p. 2, l. 6). In line 10-11 "..but under contemporary conditions they can only thrive in extreme environments that limit Metazoan competition". Given the fact that you find about 10_10 bacteria per gram forest or grassland soil, this statement does not hold or its meaning in this context here should be clarified.

Reply: This criticism can be circumvented by a condensation of the biological discourse as this is not the main thrust of this paper. However some of this may be required for the discussion.

Ref 2: Specific comments: p. 2, l. 4: "more plausible Precambrian stromatolite analog" - more plausible compared to what? p. 3, l. 25-27: What does it mean that prokaryotes dominate? What about unicellular eukaryotes in such environments? p. 4, l. 9: Which coastline, please specify. p. 4, l. 26-35: It is not clear which sites this information refers C2 BGD Interactive comment Printer-friendly version Discussion paper to. p. 8, l. 31-32: This is a rather vague statement: Why should these factors then be considered?

Interactive comment on Biogeosciences

Reply: These comments are accepted]

Discuss., https://doi.org/10.5194/bg-2017-188, 2017.

———————————————————

---

## Author Response (AR1)

The manuscript "Extant shore-platform stromatolite (SPS) assemble" by Alan Smith et al. observed SPS lithological and geomorphological assemblage and described that SPS are produced by mineral precipitation.

However, the data provided by the authors are not sufficient to support interpretations and conclusions, at least in the way they are described in the manuscript in the present form.

The present data are *definitively publishable* but not to a well-known journal as Biogeosciences.

This paper has been re-written to accommodate the referees comments. In addition new data such as the SPS localities from Luskentyre Bay, Scottish Hebrides has been added. This discovery highlights the global nature of extant SPS.

This is the only growing marine stromatolite environment that can be directly compared to an Archean marine fossil environment, ie Strelley Pool (Allwood et al. (2006). They interepreted the 3.4Ga stromatolite lower boundary as a wave platform. This makes the SPS setting very important and has been highlighted.

The paper title has been changed to reflect that this is a Geomorphological/ sedimentological/ stratigraphical investigation rather than a microbial investigation. This has/ is being done by others (Rishworth et al 2016; Edwards et al., 2017) and published papers are quoted.

The abstract has been changed to reflect this.

The Introduction will has been restructured along these lines;

> ➢ Defn of Stromatolites, ie trapped and bound and/ or mineral pptn. State that the Archean are 99% mineral pptn.

> ➢ The SPS assemblage is a new extant stromatolite marine environment.

> ➢ We have accentuated that SPS are mineral precipitated stromatolites, whereas Shark Bay and Carribean are of the trapping and binding variety. Thus the SPS setting lends itself to a direct comparison with Archean stromatolite occurrences.

> ➢ Work has been done on Cape Morgan, South Africa (Smith & Uken, 2003; Smith et al, 2005; 2011) and the Giants Causeway SPS (Cooper et al., 2013) and other SPS from the Eastern Cape, South Africa (Perissinotto et al., 2014; Rishworth et

al., 2016; 2017; Edwards et al 2017) but no attempt has been made for a detailed comparison of all the known SPS occurrences. We do this in this paper.

➢ This paper compares new SPS discoveries and all known global SPS occurrences to determine similarities and differences. From this we try to distil a SPS Facies Association. As this is work on a new stromatolite setting it has to lean heavily on fieldwork, comparisons and facies analysis.

➢ This facies analysis is then used to compare with some Archean examples, especially the 3.4 Ga palaeo SPS occurrence at Strelley Pool.

General comments

1.  The microorganisms were prevalent in the Precambrian. Microbial fossils present the cellular structure, which is similar to cyanobacteria and other prokaryotes. But morphological analysis of these microbial fossils is often not enough to obtain the correct information. [this is not required as other papers have been quoted]

2.  The acquisition of information would highly bene- fit from in-situ analytical techniques such as laser Raman spectroscopy, and stable C1 BGD Interactive comment Printer-friendly version Discussion paper isotopes. [This has not been done as it appears to be a distraction but the relevant papers are quoted. The purpose of this paper is to fully describe the SPS geomorphological setting and to indicate its global (Archean and possibly extraterrestrial) occurrence and significance. This is significant biologically but this is not a thrust of this paper.]

3.  The potential effect of microbes in the formation of stromatolites should be further discussed. [this paper discusses the geomorphology, and potential stratigraphy, consequently a microbe analysis is not part of this paper.]

4.  The scientific methods and assumptions are not valid and clearly outlined, such as when the authors try to discuss SPS preservation potential in the geological record, what is the preservation mechanism? [This can only be done here in a general way as a lot more information would be required. However there is a palaeo SPS example from the Strelley Pool (3.4Ga) Archean stromatolite assemblage and possibly the lower Gunflint Chert. If these are in fact correct interpretations then preservation (irrespective of how) is possible.]

5.  The formation of stromatolites requires certain environmental conditions, especially more information about the water chemistry measurements should be provided. [These conditions have been met here. This was not part of this project. Some work has been done and is quoted. A detailed geochem study is presently under way.]

6.  Specific comments: These have been addressed.
In this manuscript the authors compare different stromatolite assemblages along the south-east African shoreline, compare in situ factors under which they form, and evaluate their potential to gain further understanding of Precambrian stromatolite formation as well as their potential as an indicator of previous life on Mars.

While the topic in general is interesting and a comparison of recent and ancient stromatolite formation across different geological **settings may make an important contribution to our knowledge of this field**, the paper appears rather descriptive and lengthy and may benefit from some restructuring and focusing. [The paper has been more focused.]

In the first place, I would suggest to modify the title in a way that it at least contains more specific information about this study or reflects the major outcome. [This has been done and the title changed to:

**"The Extant Shore-Platform Stromatolite (SPS) Facies Association: A glimpse into the Archaean?"**

Similarly, the abstract appears as a listing of findings of this study. Here, a clear statement of the motivation of this study and highlighting the major outcome in C1 BGD Interactive comment Printer-friendly version Discussion paper one concluding sentence should be added. [This has been done.]

The motivation of this work is stated rather clearly at the end of the introduction section. However, especially in the results section, a stronger structuring along the original research objectives would help the reader find their way through this large amount of detailed site information which is provided in the results section. [This has been done.]

Although some of the site-related information is already presented in tables, comparison of key features across sites would benefit from a more condensed presentation in tables rather than in text. This way it would be easier for the reader to recognize in which key aspects the different study sites differ and what might be the most important regulating factors for stromatolite formation. This would also help the authors to check carefully if really all the aspects that they provide in the results section are needed for the discussion. [The tabulated information has been subdivided into a total of 5 separate tables so that the information can be better understood and appreciated.]

The discussion is already written quite concisely, however, a more direct reference to the objectives stated in the introduction would be helpful. For example, the discussion of the potential of SPS as Precambrian analogues remains rather superficial. [This has been bene beefed up.]

In addition, I have two concerns regarding the integration of aspects of microbiology. The authors integrated a longer section about the role of prokaryotes in stromatolite formation and the importance of the competition between prokaryotes and metazoans. However, this aspect is not targeted at all in the results section and only occasionally addressed in the discussion, and I was wondering why it was introduced so thoroughly in the introduction. [I accept this criticism and this section has been reduced. The introduction has been given a much stronger geomorphological and lithological focus. This needed to be more clearly stated at the outset as the referees are clearly expecting a strong biological focus to follow.]

In addition, some of the statements in this paragraph of the introduction (p. 2, l. 5-14) are not correct or are not sufficiently explained. What is meant by the statement that "Prokaryotes, however, do not react well to Metazoan competition"? (p. 2, l. 6). In line 10-11 "..but under contemporary conditions they can only thrive in extreme environments that limit Metazoan competition". Given the fact that you find about 10_10 bacteria per gram forest or grassland soil, this statement does not hold or its meaning in this context here should be clarified. [This has been addressed.]

Specific comments: p. 2, l. 4: "more plausible Precambrian stromatolite analog" - more plausible compared to what? p. 3, l. 25-27: What does it mean that prokaryotes dominate? What about unicellular eukaryotes in such environments? [This has been addressed.]

---

## Author Response (AR2)

**Suggestions for revision or reasons for rejection (will be published if the paper is accepted for final publication)**

5  The manuscript by Smith and colleagues has been substantially improved in its revised version. The abstract now provides a very good and well structured overview of the research topic and key findings. The results section strongly benefited from the re-arrangement of tables and from a more focused presentation of the results. All my previous comments have been addressed by the authors.

Thank you – we have attended to all comments

Specific comments:

15  p. 2, l. 22: please change to „deserve scrutiny"

p. 8, l. 6: Figure ??? needs correct figure number.

**Suggestions for revision or reasons for rejection (will be published if the paper is accepted for final publication)**

This report is very interesting and may become important for people interested in stromatolites since it offers new modern analogues which sound closer in terms of textures (although we do not see this at all in the present version of the manuscript) with Archean stromatolites. However, be careful about remaining mistakes in the manuscript. The point about tufas in the Archean seems overstated to me. I do not see why they should exist for sure in the Archean. Just present this as a possibility instead of a certainty. This suggestion has been followed.

The implications for Mars seem also very speculative and I think you should be clearer about this and better explain the connections between the sites you refer to and the ones you explored on the Earth. I detail my comments thereafter: [by its very nature this is speculative – comments follow in the following list]

- L10 : Metazoan

- « stromatolite-building organisms … thrive in extreme environments… and salt lakes » : This should be toned down regarding the extreme character of the environments. You forget lakes which are not particularly salty/hypersaline or at

least that are not called salt lakes and which are not so extreme, i.e. alkaline lakes such as those found in Mexican volcanic craters (e.g., Saghaï et al., 2015 Frontiers in Microbiology) I think that this opens up a new discussion as to whether lakes are extreme. They are low in diversity so perhaps they are

- L17: Stromatolites are not only formed by the metabolic activity of Cyanobacteria. Anoxygenic phototrophs (Bosak et al., 2007 Geobiology) and in general microorganisms feeding the alkaline engine can play as wall (e.g., Dupraz et al 2009 Earth science review) We have modified the definition

- Caption of Table 1: explain the meaning of Min Ptn done

- How do you practically affiliate a stromatolite to trapped and bound vs mineral precipitation? Could you mention the criteria? This has been done in the section that deals with the new figure 5

- P7, L9: replace X by B I guess? √

- L11: there is no Fig 3D √

- L11: how do you know that runoff is strong there? This is not really obvious from your picture. I have modified the text

- P8, L 6: replace ??? √

- P8 L10: I do not understand this thermocline. Warm water should be lighter and therefore above! I guess this is instead a chemocline here with saltier, hence denser water (and at the same time warmer) beneath. This needs to be the subject of further research but I have modified the text

- P8, L13: where can I see the peat bog? done

- Fig 5B: what am I supposed to see except big boulders? Where is tufa? Can you show a close-up I have modified the caption

- P12, L11: do not capitalize recent; What is tofu? You mean tufa? yes

- P13, L1: you refer to Fig 5 for Tinley Manor, Ballito and Cape Morgan but none of tehse sites is shown in Fig 5. Do you mean Fig 6? text is modified

- Port Edward: picture of the site? Richards Bay? neither are very photogenic, unfortuneately

- Table 4: explain meaning of abbreviations: SLR, SST done

- Discussion: what do you call tripartite stromatolite morphology? text modified

- What do you mean by climax lamination? text modified

- "The presence of tufa within the SPS facies association and its similarity to that of the 3.4 Ga Strelley Pool stromatolites strongly hints that microbial life existed in Archean terrestrial settings": I do not understand why the presence of tufa within the SPS facies hints that microbial life existed inArchean terrestrial settings. Could you elaborate? Do you mean that anytime we see stromatolites we can infer that there were tufas and hence terrestrial microbial communities? How do we know? What makes the tufas necessary to the formation of stromatolites? Moreover, are not there "abiotic" tufas? I have modified the text to explain this. If it abiogenic it would be travertine? but the terminology needs an overhaul

- Could you comment about the size of the extant stromatolites you observe and the size of Archean stromatolites? done

- P21; l26: I do not understand at all the connection between phosphorous and the presence of stromatolites on Mars. Could you elaborate? If stromatolites form by the precipitation of CaCO3, you would not expect a higher P concentration to favor CaCO3 precipitation. the text has been modified to expand on this.

- P21-22: I do not understand the connection between the marine SPS described here and possible hydrothermal deposits and/or possible lacustrine deposits. This is a bit of a stretch which is not necessary here. the hydrothermal reference has been dropped but shoreplatforms can form on any body of water with waves

.- Could you show more pictures of handsized samples and document laminations characteristic of stromatolties? I understand that such pictures may be in other papers but it would be good for the reader of the present manuscript to have some idea about how the samples look like I have inserted a new Figure 5 to try and remedy this

[revised manuscript text omitted]